# The changing epidemiology of hemorrhagic fever with renal syndrome in Southeastern China during 1963–2020: A retrospective analysis of surveillance data

Rong Zhang[1], Zhiyuan Mao[2], Jun Yang[3], Shelan Liu[1], Ying Liu[1], Shuwen Qin[1], Huaiyu Tian[4], Song Guo[1], Jiangping Ren[1], Xuguang Shi[1], Xuan Li[2], Jimin Sun[1]*, Feng Ling[1]*, Zhen Wang[1]*

1 Key Laboratory of Vaccine, Prevention and Control of Infectious Disease of Zhejiang Province, Zhejiang Provincial Center for Disease Control and Prevention, Hangzhou, China, 2 MPH department, college of Veterinary Medicine, Cornell University, Ithaca, New York, United States of America, 3 Institute for Environmental and Climate Research, Jinan University, Guangzhou, China, 4 State Key Laboratory of Remote Sensing Science, College of Global Change and Earth System Science, Beijing Normal University, Beijing, China

* jmsun@cdc.zj.cn (JS); fengl@cdc.zj.cn (FL); wangzhen@cdc.zj.cn (ZW)

**Data Availability Statement:** All relevant data are within the manuscript and its Supporting Information files.

## Abstract

### Background

Hemorrhagic fever with renal syndrome (HFRS) is a rodent-borne disease caused by hantavirus which was endemic Zhejiang Province, China. In this study, we aim to explore the changing epidemiology of HFRS in Zhejiang, identify high-risk areas and populations, and evaluate relevant policies and interventions to better improve HFRS control and prevention.

### Methods

Surveillance data on HFRS during 1963–2020 in Zhejiang Province were extracted from Zhejiang Provincial Center for Disease Control and Prevention archives and the Chinese Notifiable Disease Reporting System. The changing epidemiological characteristics of HFRS including seasonal distribution, geographical distribution, and demographic features, were analyzed using joinpoint regression, autoregressive integrated moving average model, descriptive statistical methods, and Spatio-temporal cluster analysis.

### Results

From 1963 to 2020, 114 071 HFRS cases and 1269 deaths were reported in Zhejiang Province. The incidence increased sharply from 1973 and peaked in 1986, then decreased steadily and maintained a stable incidence from 2004. HFRS cases were reported in all 11 prefecture-level cities of Zhejiang Province from 1963 to 2020. The joint region (Shengzhou, Xinchang, Tiantai, and surrounding areas), and Kaihua County are the most seriously affected regions throughout time. After 1990, the first HFRS incidence peak was in May-June, with another one from November to January. Most HFRS cases occurred in 21-

**Funding:** This study was supported by National Natural Science Funds of China (81872675), the medical research program of Zhejiang Province (2019KY358), and the state project for scientific & technological development of the 13th five-year plan in China (2017ZX10303404008002) led by JS, a grant from science technology department of Zhejiang Province (LGF20H260001) led by RZ. The funders had no role in study design, data collection and analysis, decision to publish, or preparation of the manuscript.

**Competing interests:** The authors have declared that no competing interest exist.

(26.48%) and 30- years group (24.25%) from 1991 to 2004, but 41- (25.75%) and 51-years (23.30%) had the highest proportion from 2005 to 2020. Farmers accounted for most cases (78.10%), and cases are predominantly males with a male-to-female ratio of 2.6:1. It was found that the median time from onset to diagnosis was 6.5 days (IQR 3.75–10.42), and the time from diagnosis to disease report was significantly shortened after 2011.

## Conclusions

We observed dynamic changes in the seasonal distribution, geographical distribution, and demographic features of HFRS, which should be well considered in the development of control and prevention strategies in future. Additional researches are warranted to elucidate the environmental, meteorological, and social factors associated with HFRS incidence in different decades.

### Author summary

This study conducted a long-term and systematic study on the epidemiological characteristics of HFRS in Zhejiang Province from 1963 to 2020 through a combination of time and space analysis and epidemiology, aiming to analyze the distribution characteristics of HFRS and explore the high incidence of epidemics in Zhejiang Province Regional influence. From 1963 to 2020, all 11 prefecture-level cities in Zhejiang Province reported HFRS cases, and the morbidity and mortality rates decreased significantly. However, the geographical distribution of endemic areas has been expanding to eastern Zhejiang Province. Moreover, the age of high-risk groups increases over time. Although the incidence rate has declined in recent years, HFRS is still a huge threat to people's health. As the incidence rate changes, some epidemiological characteristics have also changed. Comprehensive interventions should also be adjusted, including rodent control in endemic areas, health education, vaccination, and improved detection and diagnosis capabilities for HFRS epidemiological changes.

## Introduction

Hemorrhagic fever with renal syndrome (HFRS) is a rodent-borne illness caused by hantaviruses including Hantaan (HTNV), Seoul (SEOV), Dobrava-Belgrade virus, Saaremma, and Puumala [1]. Most HFRS patients are infected through direct exposure to the aerosolized droppings or body fluids of infected rodents, but the human-to-human transmission is rare [2]. Main clinical manifestations include fever, vomiting, abdominal pains, hypotension, kidney injury, thrombocytopenia, and shock [3].

Although suspected HFRS infection records can be traced back to the 1930s in Russia, Japan, and China, this disease did not capture modern society's attention until the Korean War in 1950 [4]. From 1950 to1953, over 3,000 United Nations soldiers acquired Korean hemorrhagic fever, which is the old name of HFRS [4]. HFRS outbreaks kept occurring in Asia and Europe in the following decades [4]. However, it remained poorly understood until 1978 when HFRS's etiological agent—Hantaan Virus and its reservoir—*Apodemus agrarius* were discovered by Lee et al [5].

HFRS infection was first observed in Japanese troops and among the locals in the Heilongjiang Province of China in the 1930s [4]. Since 1973, the HFRS epidemic in China has started expanding. In 1980, 23 provinces in China reported HFRS cases with a case fatality rate of 6.4% [4]. Then, the epidemic reached the peak in 1986, with a total of 115,804 cases reported nationwide [6]. After the government launched a series of disease control and prevention measures in 1986, the incidence began to decrease [6]. From 1990 to 2010, global annual HFRS cases were about 60,000 to 150,000, with more than 50% reported in China [7]. In the recent ten years, annual HFRS incidence was below 1 per 100,000 populations [8].

Zhejiang Province is an HFRS endemic region, and the first case was reported in 1963. It has seen a rage of HFRS outbreaks in the 1970s and 1980s. In 1984, Zhejiang Province launched the HFRS surveillance system, and the case number has started to fall. Nevertheless, currently, there are still hundreds of HFRS cases reported in Zhejiang Province annually. Zhejiang Province is located on the southeast coast of China (27°-32°N, 118°-123°E), and it occupies 101,800 square kilometers, with 74.63% being mountainous area [9]. Dominant climate in Zhejiang Province is subtropical monsoon climate, with the annual average temperature being 18 degrees Celsius and annual average precipitation being 1500mm [9]. The combination of the climate and landscape significantly contribute to the growth and reproduction of rodents, which are primary reservoirs of Hantavirus, indirectly facilitating the transmission of HFRS in Zhejiang. This study's goal is two-fold: 1) to review HFRS cases reported through surveillance systems across Zhejiang Province from 1963 to 2020 and 2) to analyze the epidemiological features of HFRS across Zhejiang Province from 1963 to 2020.

## Methods

### Ethics statement

This study was reviewed and approved by the Ethics Committee of the Zhejiang Provincial Center for Disease Control and Prevention (No.2020-021). All the data of the individuals were kept confidential as requested and ethical approve.

### Data collection and source

According to the health industry standard of the People's Republic of China for diagnostic criteria of HFRS, HFRS cases were classified as suspected cases, clinically diagnosed cases and confirmed cases. (1) A patient with epidemiological history and clinical manifestations with fever or gastrointestinal symptoms was defined as a suspected case; (2) A suspected case with hypotension, renal function impairment, increased peripheral blood cell counts and thrombocytopenia, and a positive urine protein was defined as a clinically diagnosed case; (3) A clinically diagnosed case with one or more of the following criteria: the serum specific IgM antibody is positive; the specific IgG antibody is 4 times higher than that in the acute phase; Hantavirus RNA is positive or Hantavirus is isolated was defined as a confirmed case.

The data on HFRS cases were obtained from two sources: Zhejiang Provincial Center for Disease Control and Prevention (CDC) and the Chinese Notifiable Disease Reporting System. Data from 1963 to 2004 were obtained from Zhejiang CDC archives. Due to limited surveillance capacity, data from 1963 to 1990 only included monthly county-level case number. Data from 1991 to 2004 contained monthly county-level case numbers and the demographic information of HFRS patients (age, gender, occupation). Data from 2005 to 2020 were collected from the Chinese Notifiable Disease Reporting System and contained individual-level information, including residential address, illness onset, time of diagnosis, time of death, gender, age, outcomes, and occupation.

Demographic data of each county in Zhejiang Province from 1963 to 2004 were obtained from the Comprehensive Statistical Data and Materials on 50 Years of New Zhejiang; data from 2005 to 2020 were obtained from the Zhejiang Bureau of Statistics (http://tjj.zj.gov.cn/). The base layer of the map of Zhejiang Province was supported from National Earth System Science Data Center, National Science & Technology Infrastructure of China (http://www.geodata.cn).

## Statistical analysis

To analyze the change of HFRS incidence in Zhejiang Province, we included all case data from 1963 to 2020. Incidence (per 100, 00 0 population) was defined as the number of annual new HFRS cases divided by the total population each year. We used the Joinpoint regression to examine the incidence trend from 1963 to 2020, and we described the trend with the annual percentage change (APC). To analyze the trend of HFRS, we included data from 1971 to 2010. Data before 1971 were excluded because multiple years did not receive any reports of HFRS death cases. For the same reason, we did not include data from 2011 to 2020 in our case-fatality rate (CFR) analysis. We define the CFR (per 100 population) as the number of death cases divided by the number of clinical cases and lab-confirmed cases each year. The student $t$-test was applied to assess if the APC significantly differs from 0; the significance level of a two-sided test was set as 0.05. We described the trend as "increase", "decrease", or "stable" based on the $t$-test results. "stable" means a non-significant APC ($p > 0.05$); both "increase" and "decrease" relate to a significant APC ($p < 0.05$)). The parametric method was employed to estimate the 95% Confidence Intervals for APC and the average annual percentage change (AAPC) for specific periods, assuming APC intervals follow a t distribution and AAPC intervals following a normal distribution. All joinpoint regression analysis was conducted with the Joinpoint software (Version 4.8.0.1)

To study the change of surveillance capability and patients' medical seeking behavior, we estimated the distribution of time from disease onset to diagnosis, time of diagnosis to death between different period, and between different diagnosis type. Also, we conduct logistic regression to study factors that may affect case confirmation (clinical diagnosis vs. laboratory confirmation), with the backward selection of demographic variables (age, gender, occupation), geographical variables (City), year of illness, and time interval from disease onset to diagnosis.

The epidemiological characteristics of HFRS cases, including geographic distribution, seasonal pattern, gender, age, and occupation, were analyzed using descriptive methods. As no related data were obtained, the descriptive analysis was not conducted for patients from 1963 to 1990. Our study combined the data's characteristics and divided the data into the following four stages: 1963–1979, 1980–1989, 1990–2004, and 2005–2020. We employed Fisher's exact test to assess the CFR difference by age, gender, and patient occupation, respectively. The difference was considered statistically significant when $p$-value < 0.05.

As for the time series analysis of HFRS, we created heat maps of the monthly HFRS cases 1973–2020; analysis for 1963–1972 were not conducted due to lack of relevant data. Then, we fitted the dataset with seasonal autoregressive integrated moving average models (SARIMA). For prediction validation, we included data from 2005 to 2019 as the training and made data of 2020 as the holdout set. Through Box-Jenkins Tests and the goodness-of-fit tests, the optimal SARIMA (1,0,0) (1,1,2) with drift was identified. Finally, we predicted the HFRS case number in Zhejiang Province in the coming 12 months. The logistic regression, Fisher's exact test, and SARIMA analysis were completed by R software (Version 4.0.1)

To study the impact of policies and interventions on county-level HFRS epidemics, we employed SaTScan (http://www.satscan.org/) software (version 9.6) to perform a space-time

analysis for counties before and after the changes of policies and interventions, using discrete Poisson models [10]. We set the national rodent-control campaign (starting from 1986) and the Zhejiang Provincial HFRS vaccination campaign (starting from 1995) as our targets; the HFRS case data from 1985–1987, 1994–1996 were included in the analysis A high-risk cluster was determined as if a geographical area contained significantly more HFRS cases than the provincial average, during a specific period. In the null hypothesis, we assumed the relative risk of any area in Zhejiang to be 1. For space-time analyses, spatial window set maximum spatial cluster size 50% of the population at risk, temporal window set minimum temporal cluster size two months and maximum temporal cluster size 50% of the study period. Cluster restrictions set a minimum number of cases in at least 2 cases. The inference number of replications was 999.

## Results

A total of 114 071 cases and 1269 deaths were reported from 1963 to 2020 in Zhejiang Province. The annual average number is 1978, with the highest number of 10 950 recorded in 1986. The incidence rate fluctuated over the past 50 years (Fig 1). A total of four joinpoints were identified via our final selected joinpoint regression models (joinpoint year: 1966, 1969, 1973, and 1986). Therefore, we divided the overall trend of HFRS incidence from 1963 to 2020 into five segments based on the joinpoints (Fig A in S1 Fig). From 1963 to 1966, the incidence rate had kept stable at the level of 0.05 per 100,000 ($t = 1.5$; $p = 0.1$); from 1966 to 1969, the incidence rate decreased, with an APC of -59% ($t = -2.5$; $p<0.01$)); after 1969, the HFRS incidence rate exponentially increased, with an APC of 305.9% ($t = 7.7$; $p<0.01$), and reached the rate of 1.35 per 100,000 in 1973. In the next 13 years, the HFRS incidence rate kept growing with an APC of 23.2% per year and reached the peak of 27.04 per 100,000 in 1986 ($t = 9.7$; $p<0.01$). After the Chinese government stipulated a series of control measures in 1986, the incidence rate has begun decreasing, with an APC of -11.2% ($t = -26.5$; $p<0.01$).

Although the number of HFRS cases in Zhejiang Province has been steadily decreasing over the past 30 years, the spatial distribution of HFRS infection has been expanding in recent years (Fig 2). The joint region (Shengzhou County, Xinchang County, and Tiantai County) was the epicenter of HFRS in Zhejiang, with over 32,000 cases recorded over the past fifty years. And Zhejiang's western areas, including Jiande County, Kaihua County, and the surrounding areas, was another epicenter in Zhejiang Province with over 12,000 cases reported. Of note, the HFRS epidemic in Zhejiang's western region has been considerably alleviated in recent 30 years, and most high-incidence areas clustered are currently in the eastern region (except for Kaihua County and Longquan County in the southwest).

The retrospective space-time analysis showed that, compared to the number of HFRS clusters in 1985–1986, the number substantially decreased in 1987–1988 and the HFRS clusters in the northeastern region and south region have disappeared. However, emerging HFRS epidemics in western Zhejiang Province were detected in 1978–1988, while no clusters were detected in the same region during 1985–1986. Also, for 1996–1997, the number of HFRS clusters was significantly less than the number in 1994–1995, and the HFRS epidemics in eastern and central regions of Zhejiang Province were mitigated. In both comparisons, the joint region (Shengzhou County, Xinchang County, and Tiantai County) continued to be the high-risk clusters (Fig 3).

Overall, the HFRS epidemic of Zhejiang Province did not clearly show a semi-annual seasonal pattern until 1990 (Fig 4). Before 1990, there was the only peak of HFRS incidence annual, and it was during the wintertime (November to January). From 1990, another peak was observed, which was during May and June. At the municipal level, only Huzhou City,

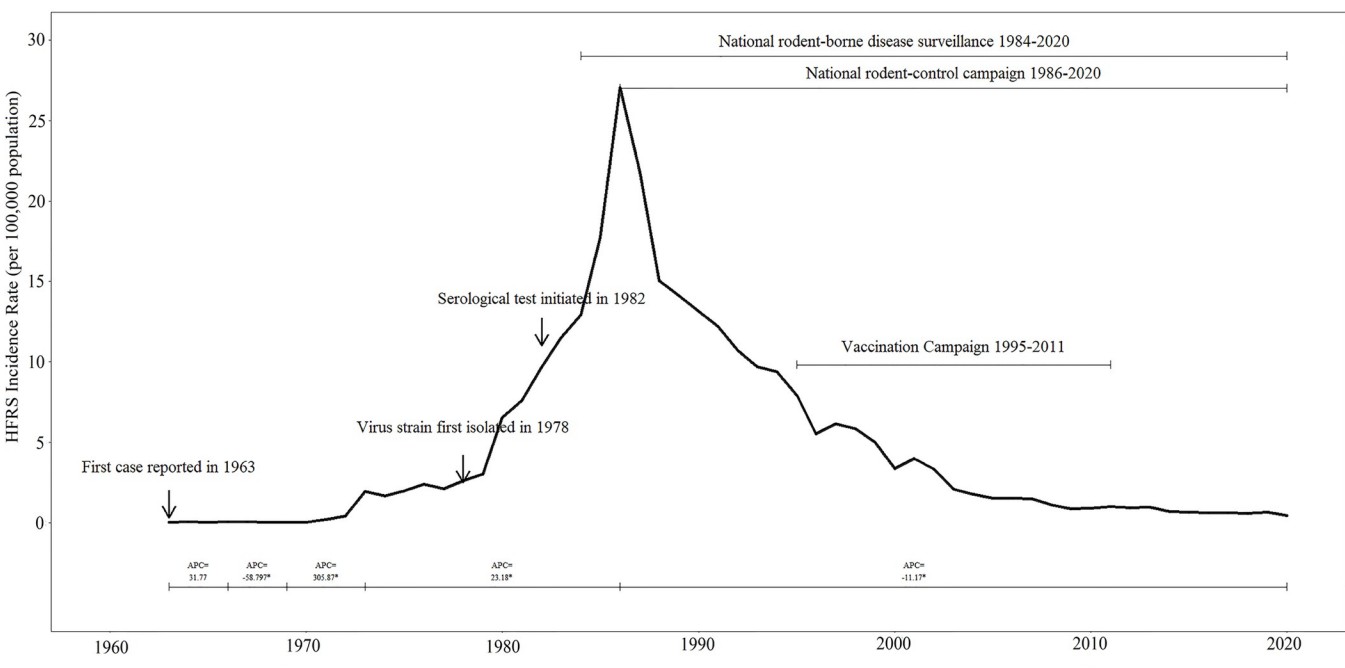

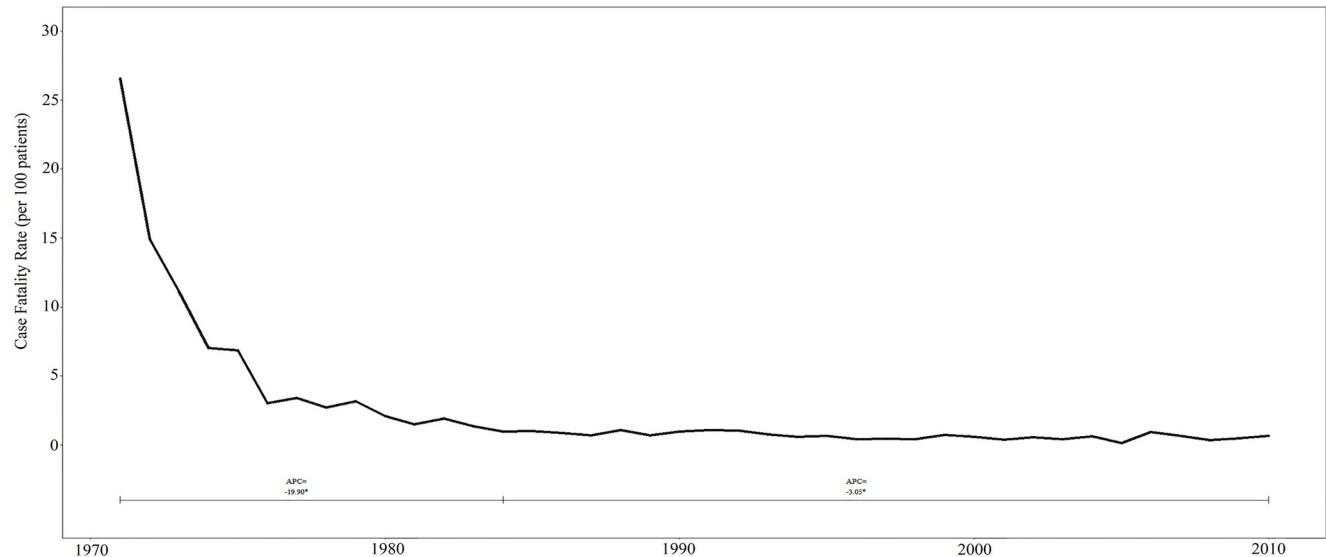

**Fig 1. The incidence and case-fatality rate of HFRS in Zhejiang Province, 1963–2020, 1971–2010.**

Jiaxing City, and Ningbo City showed a semi-annual pattern before 1990, and the other eight cities had a peak during the winter season (Figs B, C and D in S1 Fig). From 1991, most cities in Zhejiang Province, except for Zhoushan City, started to show a similar seasonal pattern, with one peak during winter and another between May and June. In our SARIMA model, the predicted values for 2020 closely matched the real data that all the actual values are within the 95% confidence interval (Figs E and F in S1 Fig).

The data on medical seeking behavior and disease reporting is only available for diagnosis after 2004. We divided the diagnosis data after 2004 into two groups: one from 2005 to 2010 and another from 2011 to 2020. The median value of the time interval from diagnosis to report

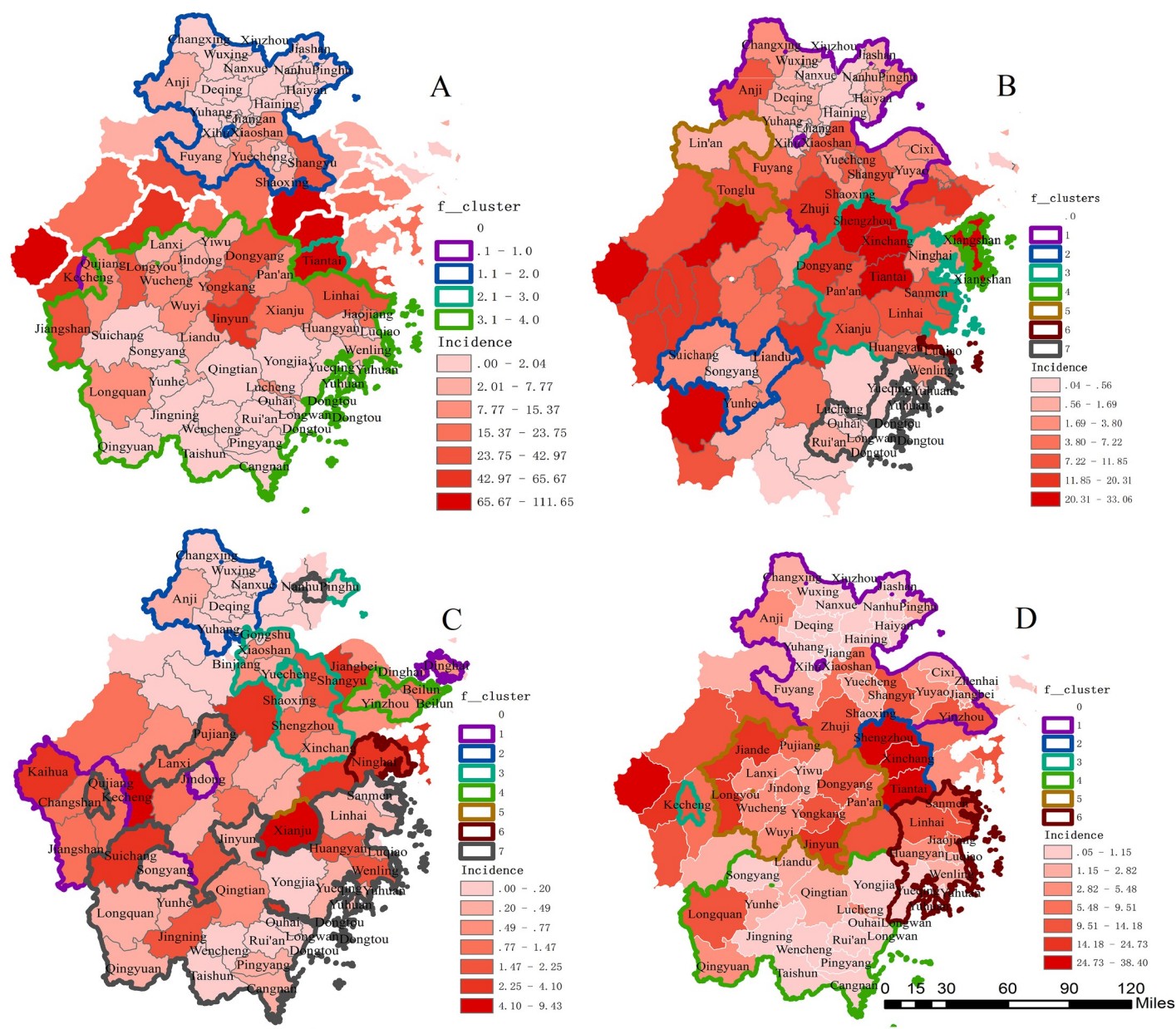

**Fig 2.** The spatio-temporal clusters overlaid with the average annual incidence of HFRS between 1980–1990 (A), 1991–2004 (B), 2005–2020 (C) and 1980–2020 (D) in Zhejiang Province, China.

for the 2005–2010 group is significantly larger than the 2011–2020 group (12 hours, IQR:11.2~32 *vs.* 4.5 hours, IQR:1.1~16.8; $W$ = 11157782, P<0.05, Fig 5). No difference was detected between the two groups, in the interval from disease onset to diagnosis, from onset to death, and from diagnosis to death (Fig 5). The median value for the onset-to-diagnosis interval from 2005 to 2020 was 6.5 days (IQR:3.75~10.42). Only 25 death cases were reported during this period, and the median value for onset-to-death interval was five days (IQR: 3~12). Regarding disease onset to diagnosis interval, onset to death interval, and diagnosis to reporting interval, no significant difference was observed between lab-confirmed cases and clinical

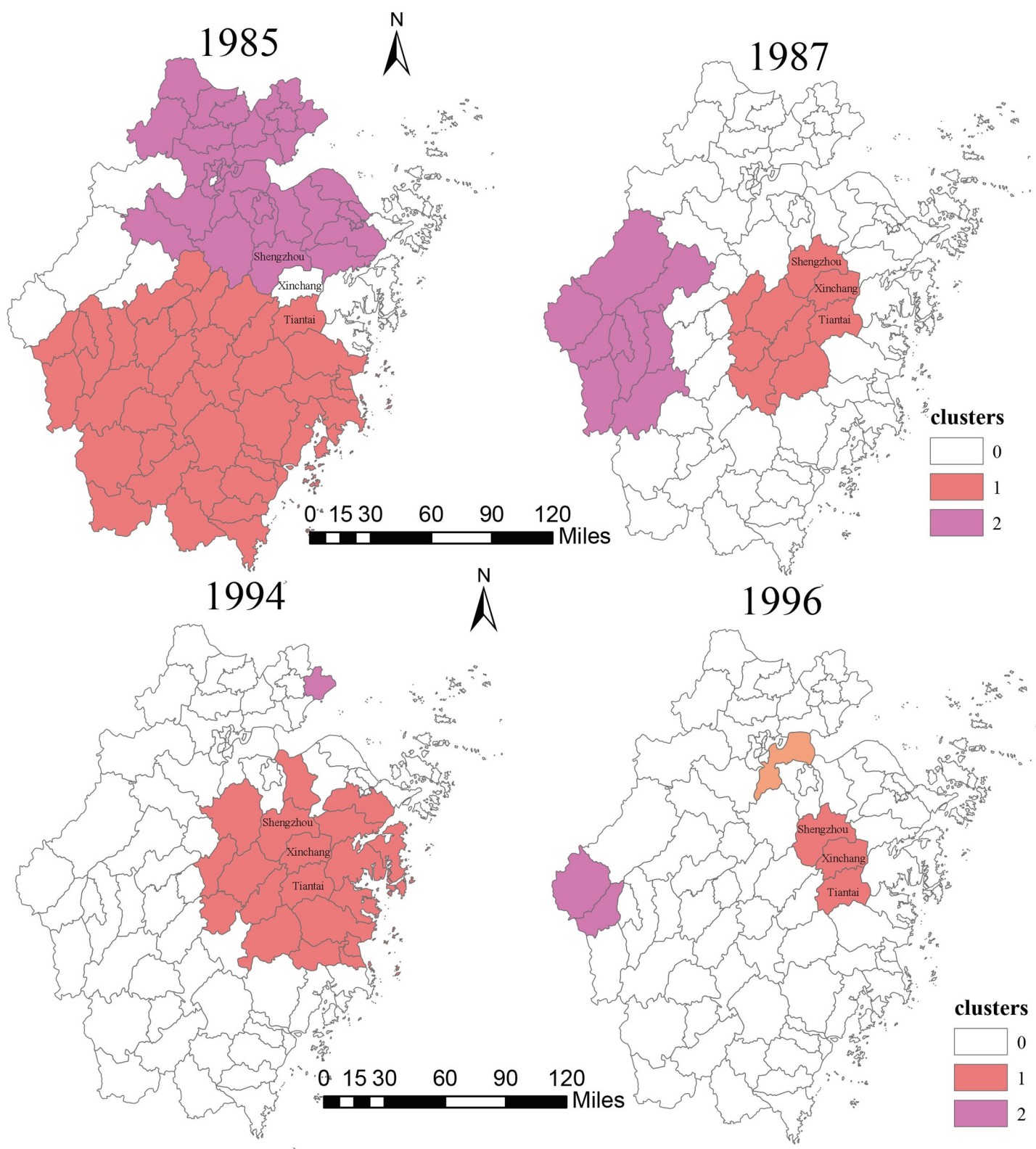

**Fig 3. Cluster of HFRS in 1985, 1987, 1994, and 1996.** Rodent control measures were conducted during 1985–1987 and vaccination was conducted during 1994–1996.

**Fig 4. Heatmap of monthly HFRS cases in Zhejiang Province, 1973–2020.**

cases (Fig 5). We identified year, city, and gender with the multivariate logistic regression as the factors influencing the case confirmation status (Table 1).

A total of 45 635 cases with 287 deaths were reported from 1991 to 2020 in Zhejiang Province. Among these cases, the overall male-to-female ratio was 2.6:1, and the median age was 46 years (IQR: 36–57). Of note, the age distribution varied greatly by time and gender (Fig 6). Before 2005, most cases were identified in both male and female patients aged from 31 to 40 years; since 2005, most cases were male patients aged 41–50 years and female patients aged 51–60 years. Besides, we discovered that farmers accounted for 78.10% of total HFRS cases, followed by factory workers (8.40%), students (3.73%), and houseworkers (1.95%, Table A in S1 Text).

The overall case fatality rate (CFR) of male cases was slightly lower than that of females (0.61% vs 0.69%, Table B in S1 Text). Also, we discovered that the risk of dying by HFRS increases as age grew for both male and female groups. The overall risk of death was significantly higher in patients over 41 years old than the youth (OR = 1.98, 95% CI: 1.56~2.53, $p$<0.001). The lowest CFR of male cases was observed in the 11–20 years group, and the highest CFR was observed in those over 70 years; the lowest CFR of female cases was observed in the <10 years group, and the highest CFR was observed in 61- years group. Still, no significant difference by gender was detected across age groups.

## Discussion

Our study has explored the incidence, mortality, geographical distribution, seasonal pattern, demographic features of HFRS cases in Zhejiang Province across the past fifty years. Also, we analyzed the impact of relevant policies and control measures made by governments and health institutions on the HFRS epidemic. Including all the HFRS cases in Zhejiang Province since the establishment of the People's Republic of China, our study is the largest-scale epidemiology study on HFRS globally.

Before 1969, in the first five years after HFRS was first recorded in Zhejiang, the HFRS incidence had shown a slight increase, followed by a drastic decrease. From 1969 to 1986, the incidence significantly increased and reached a peak in 1986. This trend closely matches the national HFRS incidence trend at the same period that the national trend also fluctuated in the 1960s and then increased to the highest incidence in 1986 [6]. China established 48 monitoring sites for national HFRS surveillance in 1984, and Tiantai County in Zhejiang Province was selected as one site [11]. The establishment of monitoring sites could partly explain the large discrepancy between observation and expectation, in terms of case incidence, from 1984 to

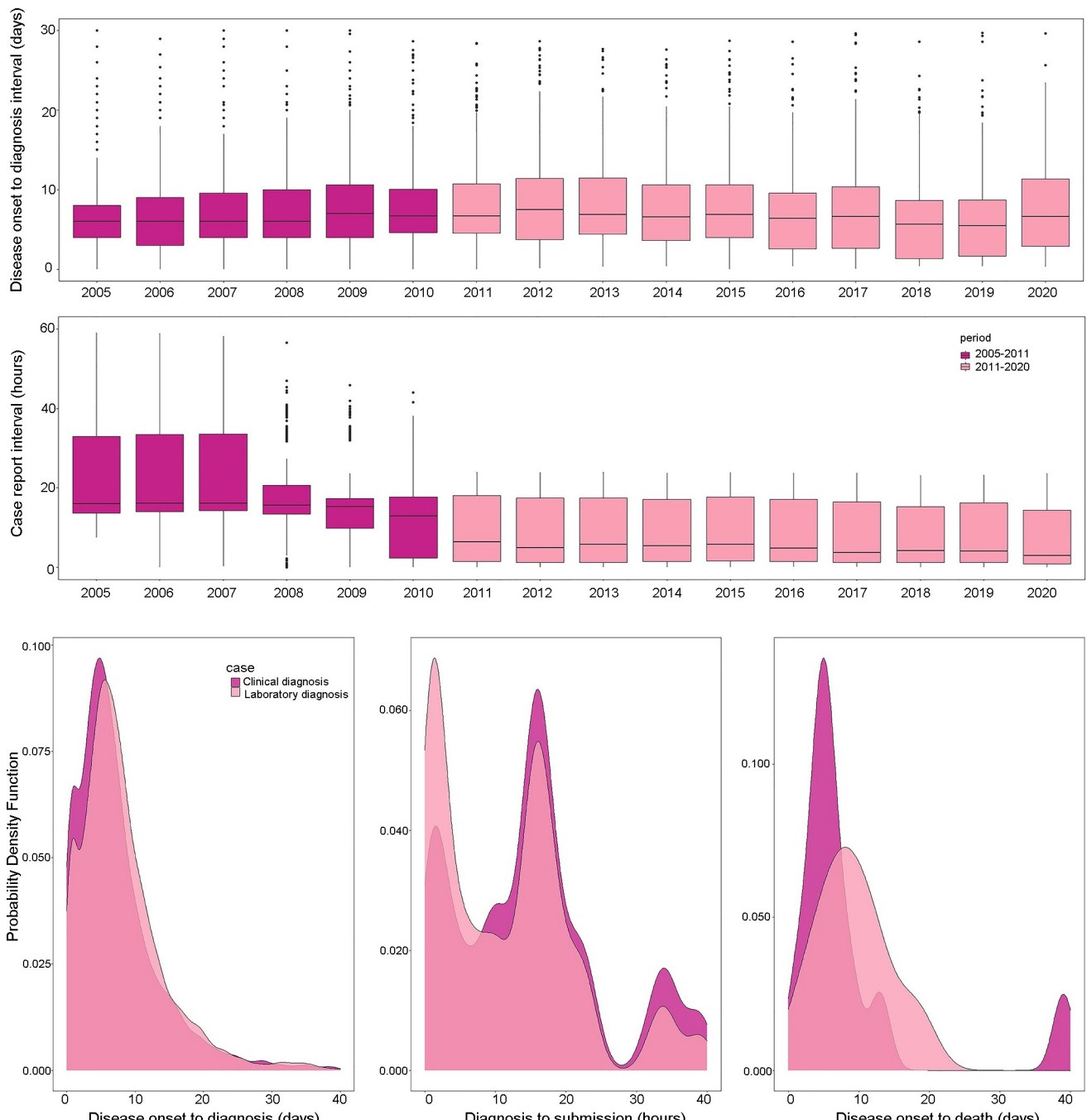

**Fig 5. Intervals from illness onset to diagnosis and intervals from diagnosis to report of HFRS cases in Zhejiang Province, 2005–2020.**

1986. The disease reporting system, diagnostic techniques, and many health workers' knowledge about HFRS infection were limited before 1984, so the actual number of annual HFRS infections may be considerably larger than the reported number. However, the expected values were calculated based on the surveillance capability before 1984. As the national rodent-control campaign was launched in 1986, Zhejiang's HFRS incidence decreased consistently and

**Table 1. Factors associated with case type and demographic information about death cases, 2005–2020.**

| | | Case type | | | Deaths | |
| --- | --- | --- | --- | --- | --- | --- |
| | | (Lab-confirmed: clinical) | | | | |
| | | Case Counts | Adjusted OR (95% CI) | | N = 25 | |
| Gender | | | | Gender | | |
| | Female | 1322:746 | Ref | | Female | 11 |
| | Male | 3515:2141 | 0.86 (0.77, 0.96) | | Male | 14 |
| City | | | | City | | |
| | Hangzhou | 240:110 | Ref | | Lishui | 7 |
| | Huzhou | 102:60 | 0.73 (0.49, 1.10) | | Taizhou | 5 |
| | Jiaxin | 21:5 | 1.75 (0.67, 5.43) | | Shaoxing | 4 |
| | Jinhua | 267:353 | 0.32 (0.24, 0.43) | | Ningbo | 3 |
| | Lishui | 542:351 | 0.76 (0.58, 0.99) | | Quzhou | 2 |
| | Ningbo | 1485:390 | 1.86 (1.43, 2.40) | | Wenzhou | 2 |
| | Quzhou | 286:513 | 0.23 (0.17, 0.30) | | Hangzhou | 1 |
| | Shaoxing | 709:414 | 0.87 (0.66, 1.13) | | Jinhua | 1 |
| | Taizhou | 1059:583 | 0.81 (0.62, 1.04) | Occupation | | |
| | Wenzhou | 115:107 | 0.38 (0.27, 0.55) | | Farmer | 18 |
| | Zhoushan | 11:01 | 3.93 (0.73, 72.01) | | Worker | 5 |
| Age | | | | | Freelancer | 1 |
| | 0–10 | 9:09 | Ref | | Government | 1 |
| | 11–20 | 159:84 | 2.16 (0.74, 6.26) | Age | | |
| | 21–30 | 544:253 | 2.08 (0.73, 5.86) | | 11–20 | 1 |
| | 31–40 | 937:555 | 1.92 (0.67, 5.37) | | 21–30 | 1 |
| | 41–50 | 1229:725 | 1.74 (0.61, 4.87) | | 31–40 | 4 |
| | 51–60 | 1101:705 | 1.65 (0.58, 4.62) | | 41–50 | 7 |
| | 61–70 | 581:367 | 1.55 (0.54, 4.36) | | 51–60 | 7 |
| | 71- | 277:189 | 1.43 (0.49, 4.03) | | 61–70 | 3 |
| Year | | | | | 71- | 2 |
| | 2005–2010 | 1983:1768 | Ref | | | |
| | 2011–2020 | 2854:1119 | 2.75 (2.48, 3.04) | | | |

steadily. The national rodent-control campaign contains a series of rodent-control measures, which would be applied to both urban and rural areas in spring and fall.

Our results are consistent with the findings from previous studies [11,12]. We found that there were more male HFRS patients across all age groups in Zhejiang Province, with the overall gender ratio of male-to-female being 2.6:1, which is close to the ratio of 3:1 in Zibo City, another eastern coastal city of China. Despite the difference of case numbers by gender, the case distribution was "aging" for both gender. In the period of 1991–2004, most patients belonged to the age group 31–40 years for both gender; in 2005–2020, the largest proportion of female patients was 51–60 years old while most male patients were 41–50 years old. Also, the proportion of patients over 60 was significantly enlarged. This "aging" phenomenon is in line with the finding of He et al [13]. This could be explained by the phenomenon of "hollow villages" in China [14]. Now, more and more young people in rural areas leave and work in cities to pursue better working opportunities, while older adults tend to stay in villages to keep farming. So, older adults are more likely to acquire HFRS infection during agricultural activities. Also, the difference in incidence by gender may be caused by the gender difference in daily activities. Males have higher opportunity to work in farmlands, favoring a higher chance of

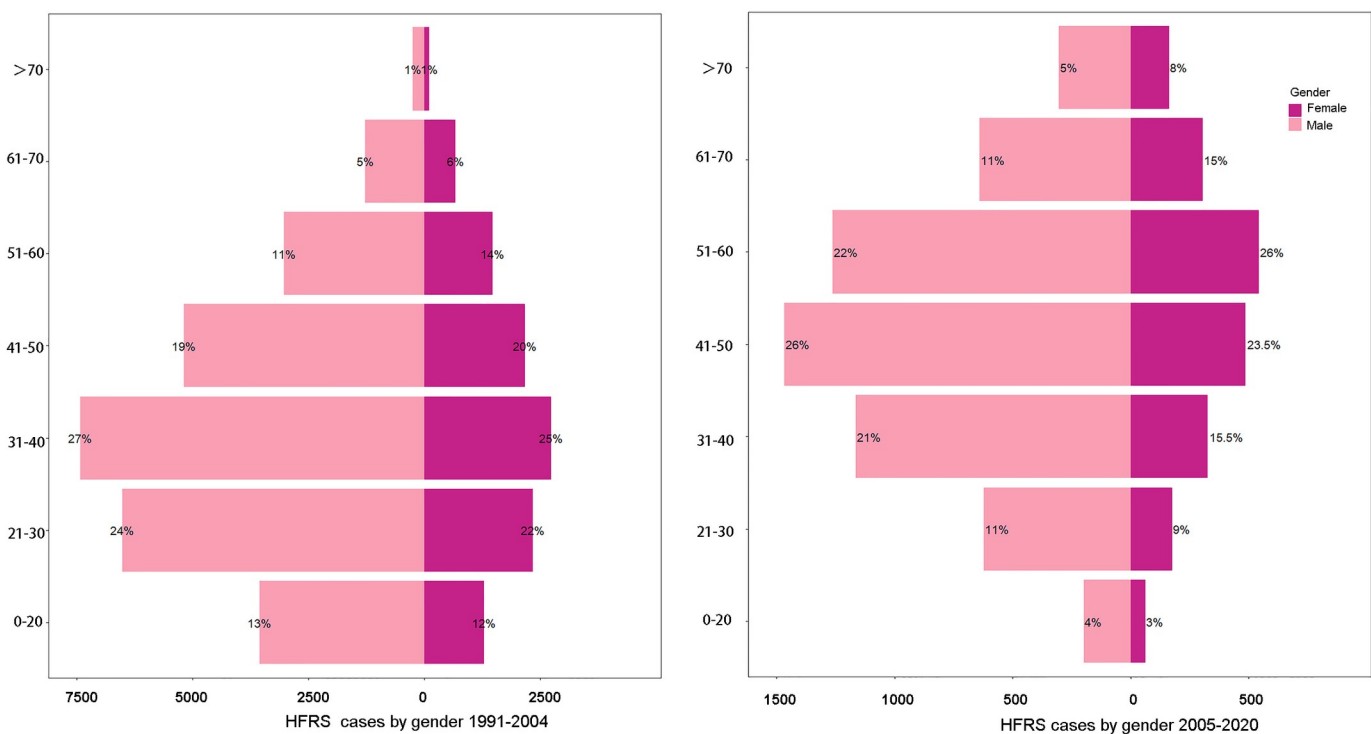

**Fig 6. The gender distribution of HFRS cases in Zhejiang Province in different age groups from 2005 to 2020.**

contacting rodents, while females often stay at home and take care of families [15]. Another study also reported that there were more male than female among imported HFRS cases [16].

Another finding from the current study is that farmers and workers are the two most common occupations contracting HFRS, which is consistent with findings from previous studies in other regions of China and Korea [17–20]. This occupation distribution could be partly explained by the fact that both farmers and essential workers are engaged with frequent outdoor activities and therefore have higher chances contacting with rodents or their body fluids containing hantaviruses. Fullhorst et al pointed out the importance of personal protective equipment (PPE) for people who have close physical contact with rodents regarding protection from hantaviruses. However, regardless of countries, the prevalence of PPE usage or other protective behaviors in agricultural activities is always dissatisfying [21,22]. Many farmers and workers are reluctant to take protective measures because they think those are burdensome and costly [23].

To assuage the health burden of HFRS, the Zhejiang government has initiated a series of research on HFRS vaccines in 1993. The overall efficacy of vaccination was 93.77~97.61%, with booster shots completed [24]. In 1995, the vaccination program started in high-risk areas of Zhejiang Province, and the lowering incidence was subsequently observed in the following year.

As shown in our geographical distribution analysis, Wenzhou City and Zhoushan City incidence remained relatively low compared to other cities within the province. This phenomenon has been observed in other coastal cities in Southern China, and the year-round surrounding rainfall and humility and association with HFRS will require further analysis [25]. Even though the rodent density in all cities in Zhejiang Province has been significantly lower down after 1986 and other prevention measures took place across the province, the joint region

(Shengzhou County, Xinchang County, and Tiantai County), Longquan County, and Kaihua County consistently became the high-risk cluster for HFRS infection throughout time. The consistency can be partly explained by the fact that these regions are entirely mountainous, and many residents here are farmers. It suggests that more comprehensive HFRS educational programs aiming at farmers and essential workers are still needed in this area.

Our space-time permutation model confirmed the overall positive effect of the rodent-control campaign and provincial vaccination campaign on controlling HFRS in Zhejiang Province. After either campaign was launched, the amount of and geographical distribution of county-level HFRS clusters were largely minimized in the succeeding year. It seems that the emergences of HFRS clusters in western Zhejiang Province after campaigns in 1987 and 1996 were conflicted with our conclusion. However, given that western Zhejiang Province was continuously a high-risk area of HFRS, we speculated that the temporary enhancement of local HFRS surveillance capability may lead to this conflict, since the HFRS-related campaigns would drive local medical workers to pay more attention on detecting HFRS cases during the campaign period.

Our finding concerning seasonal patterns of HFRS was in agreement with a previous study in Shandong Province of China that HFRS incidence only had a significant peak in winter after the introduction of the HFRS epidemic, and it took decades for the epidemic to develop into a semi-annual pattern [25]. The seasonal pattern transition represents a shift of dominant Hantavirus strains in Zhejiang Province because HTNV-caused infection tends to have the highest incidence in winter. In contrast, the SEOV-caused infection is featured with a significant outbreak in spring [26]. Also, this shift of dominant Hantaviruses was observed in other regions of China at the same time, including Shandong, Beijing, Inner Mongolia Autonomous Region [25], and it parallels with the speculation draw by the decrease in mortality rate [6]. Besides, the transition is associated with the rapid urbanization within Zhejiang, as the host of HTNV—*Apodemus agrarius* is more likely to inhabit rural areas, while the *Rattus norvegicus*, host of SEOV, is more common in urban [26].

Other than the year of disease onset, we also identified that HFRS infections in Hangzhou City and Ningbo City are more likely to be confirmed through laboratory tests than in Jinhua City, Lishui City, Quzhou City, and Wenzhou City. We speculated that this might relate to the inequity of medical resources between cities. A previous study pointed out that significant health inequity between rural and urban residents existed in Zhejiang Province [27]. Rural residents enjoyed less welfare from the current medical welfare systems, have fewer medical resources, and usually had worse outcomes than urban residents. Compared to Jinhua City, Lishui City, Quzhou City, and Wenzhou City, Hangzhou City and Ningbo City are more advanced in urbanization and have relatively abundant medical resources.

In the 1960s, only 75 cases and 12 deaths by HFRS were recorded, with an overall CFR of 16 per 100 patients. However, given the limited diagnostic capability in the 1960s, the actual case number (denominator of CFR) should far exceed 75, and the ratio is supposed to be inaccurate. From 1971 to 2010 (except for 1998), each year has at least one death by HFRS reported, and the CFR consistently decreased. After 1984, the annual CFR was less than 1. The national CFR for HFRS patients similarly decreased over time but at a slower pace [6]. The high CFR in the 1960s and 1970s should be attributed to the limited knowledge about HFRS diagnostic, prevention, and treatment at that period. With the accumulation of such knowledge and improvements in medical technology, the death rate correspondingly decreased. Besides, Zhang et al interestingly linked the decrease in CFR to the transition of disease structure (dominate virus strains) within the province [6]. The clinical symptoms and outcomes of SEOV-caused HFRS infection are usually milder than those caused by HTNV-caused HFRS infection,

which means less death [28]. Thus, he proposed that the decreasing CFR is associated with the increasing SEOV cases [6].

The difference of CFR by gender is intuitively at odds with the difference of incidence. Although the male cases were nearly three times as many as the female cases, no significant difference was detected for the overall CFR of female and male patients, which aligns with Klein et al findings [12,29].

The current study has several limitations. First, the HFRS case record used might be incomplete before 2005. The epidemiological data record requirement was standardized in 2005 when the Chinese government established the Notifiable Disease Reporting System (NDRS). Certain reports before 2005 would inevitably contain inaccurate information due to a lack of standardized procedures. For example, this might lead to the actual case number before 2005 being larger than the reported cases. Second, potential confounding effects exist in data analysis. The global pandemic greatly influences the HFRS epidemiology. However, the SARIMA model prediction was based entirely on past data, and thus, it failed to consider the current change. Therefore, there are great chances that the current prediction exceeds the actual number. Third, due to lack of relevant data, our study did not distinguish HTNV strain-type and SEOV strain-type cases. Therefore, our inference about the shift of HFRS virus in Zhejiang Province cannot be confirmed without laboratory evidence.

## Conclusion

In spite of the continuous decline of HFRS morbidity and mortality in recent years, the HFRS epidemic kept geographically expanding. Also, while farmers and essential workers are consistently at high risk of HFRS infection, this population's average age keeps increasing. More HFRS-related health promotion should target this population within Zhejiang Province. We expect this study will provide a better understanding of HFRS infection within Zhejiang Province in the past 50 years, and it also could provide the basis and direction for future research.

## Supporting information

**S1 Fig.** Fig A The joinpoint regression for HFRS incidence in Zhejiang Province, 1963–2020; Fig B Monthly average HFRS cases by cities in Zhejiang Province, 1973–1990; Fig C Monthly average HFRS cases by cities in Zhejiang Province, 1991–2004; Fig D Monthly average HFRS cases by cities in Zhejiang Province, 2005–2020; Fig E Seasonal decomposition for monthly HFRS in Zhejiang Province, 1963–2020; Fig F The predicted values of HFRS cases in Zhejiang Province for 2020 using SARIMA model.
(TIF)

**S1 Text.** Table A Estimates of HFRS patients' occupation in Zhejiang Province, 1991–2020; Table B Age distribution of HFRS cases in Zhejiang Province, 1991–2020.
(DOC)

## Acknowledgments

I would like to thank the Zhejiang Provincial and County Centers for Disease Control and Prevention for their assistance in data collection.

## Author Contributions

**Methodology:** Rong Zhang, Zhiyuan Mao.

**Project administration:** Rong Zhang, Jimin Sun.

**Resources:** Ying Liu, Shuwen Qin, Xuguang Shi.

**Software:** Rong Zhang, Zhiyuan Mao, Xuan Li.

**Supervision:** Jimin Sun, Feng Ling, Zhen Wang.

**Visualization:** Zhiyuan Mao, Song Guo, Jiangping Ren.

**Writing – original draft:** Rong Zhang, Jimin Sun.

**Writing – review & editing:** Jun Yang, Shelan Liu, Huaiyu Tian, Jimin Sun.

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
