## [Decision Letter · Decision Letter 0]

22 Jun 2021

Dear Mr. Sun,

Thank you very much for submitting your manuscript "The changing epidemiology of hemorrhagic fever with renal syndrome in Southeastern China during 1963-2020: a retrospective analysis of surveillance data" for consideration at PLOS Neglected Tropical Diseases. As with all papers reviewed by the journal, your manuscript was reviewed by members of the editorial board and by several independent reviewers. The reviewers appreciated the attention to an important topic. Based on the reviews, we are likely to accept this manuscript for publication, providing that you modify the manuscript according to the review recommendations. 

Sincerely,

Ran Wang, Ph.D., M.D.

Associate Editor

David Harley

Deputy Editor

Reviewer's Responses to Questions

**Key Review Criteria Required for Acceptance?**

**Methods**

-Are the objectives of the study clearly articulated with a clear testable hypothesis stated?

-Is the study design appropriate to address the stated objectives?

-Is the population clearly described and appropriate for the hypothesis being tested?

-Is the sample size sufficient to ensure adequate power to address the hypothesis being tested?

-Were correct statistical analysis used to support conclusions?

-Are there concerns about ethical or regulatory requirements being met?

Reviewer #1: What is important in this paper is massive epidemiological surveillance data on HFRS during 60 years in Zhejiang Province. Authors have to show ethical approve (approval No.) about patient’s specimens. Also, It is unclear by what criteria the clinical HFRS case was classified and lab diagnose was conducted (the ‘Diagnose criteria’ in ‘Methods’ section is not enough to explain). More detailed explanations were needed.

Minor comment:

“Besides, this patient was tested negative for relevant lab tests or was not tested.”

The meaning of this sentence is unclear. What kind of person does 'this patient' refer to? If this patient are the people have clinical symptoms, the meaning of sentence is vague.

Reviewer #2: -Are the objectives of the study clearly articulated with a clear testable hypothesis stated?

 Yes

-Is the study design appropriate to address the stated objectives?

Yes

-Is the population clearly described and appropriate for the hypothesis being tested?

Yes

-Is the sample size sufficient to ensure adequate power to address the hypothesis being tested?

Yes.

-Were correct statistical analysis used to support conclusions?

Yes

-Are there concerns about ethical or regulatory requirements being met?

No

Reviewer #3: methods employed are state-of the art;

- Page 1: please, change the name Dobrava to Dobrava-Belgrade virus

- Page 2: please, include the reference for the sentence: “In the recent ten years, annual HFRS incidence was below 1 per 100,000 population.”

- Page 2: in the sentence “Data from 1963 to 1990 and data from 1991 to 2004 were obtained from Zhejiang CDC archives.” There is no need to split the time period, merge to “from 1963 to 2004”; the explanation is in the following sentence and this one is just confusing

- Please, when referring to campaigns and policy changes, include brief explanation in addition title, e.g what is “Two Mountains Theory” (page 3)?

**Results**

-Does the analysis presented match the analysis plan?

-Are the results clearly and completely presented?

-Are the figures (Tables, Images) of sufficient quality for clarity?

Reviewer #1: : This is a good research of the HFRS case from 1963 to 2020 showing epidemiology according to various patterns through spatio-temporary analysis. Epidemiological trends, geographic distributions, and dynamic change of seasonal distributions are well analyzed within a Province. Notation of numbers or English needs to be improved.

Minor comment

- In Figure 3, the name of joint region (Shengzhou county, Xinchang county, and Tiantai County) must be written in figure. It is difficult for readers to intuitively identify the area.

Reviewer #2: everything is ok.

Reviewer #3: the results could be presented in a more clear way:

**Conclusions**

-Are the conclusions supported by the data presented?

-Are the limitations of analysis clearly described?

-Do the authors discuss how these data can be helpful to advance our understanding of the topic under study?

-Is public health relevance addressed?

Reviewer #1: : The conclusion is mostly make sense and acceptable, but the connection with climate seems to be overinterpreted. The year-round surrounding rainfall and humility and association with HFRS will require further analysis of the direct correlation.

Minor comment

- Virus name must be corrected. (HTNV)

- The reference style shall be unified.

- Line number must also be filled in.

Reviewer #2: -Are the conclusions supported by the data presented?

Yes.

-Are the limitations of analysis clearly described?

Yes

-Do the authors discuss how these data can be helpful to advance our understanding of the topic under study?

Yes

-Is public health relevance addressed?

Yes

Reviewer #3: - Please, present results in a more clear way; figure captions need to be more self-explanatory, especially Figures 3, 5, 6.

- Please, consider presenting joinpoint analysis results in the main text, albeit in a more structured way than currently, whereas Tables1, 3, 4 could be moved to the supplementary material

**Editorial and Data Presentation Modifications?**

Reviewer #1: (No Response)

Reviewer #2: Dear editor

Thanks on thinking of me to serve as e reviewer for this paper.

The paper is mostly informative. It is well written. It isn’t to long, so the readers can read it.

I suggest to shortness the abstract section. It’s too long. There are enough or adequate tables and figures.

Instead of the world “sex” is better to use “gender”

In the discussion section the authors try to explain the gender differences. In this paragraph I suggest to add this reference:

 Puca E, Pipero P, Harxhi A, Abazaj E, Gega A, Puca E, Akshija I (2018) The role of gender in the prevalence of human leptospirosis in Albania. J Infect Dev Ctries 12:150-155. doi: 10.3855/jidc.9805

 The authors are correct with citation of the study limitations.

I suggest to add and this reference in their paper

Puca E, Qato M, Pipero P, Akshija I, Kote M, Kraja D. Two cases of imported hemorrhagic fever with renal syndrome and systematic review of literature. Travel Med Infect Dis. 2019 Mar-Apr;28:86-90. doi: 10.1016/j.tmaid.2018.07.010. Epub 2018 Aug 13. PMID: 30114480.

I prefer to accept this paper with minor revision.

Reviewer #3: (No Response)

**Summary and General Comments**

Reviewer #1: (No Response)

Reviewer #2: Dear editor

Thanks on thinking of me to serve as e reviewer for this paper.

The paper is mostly informative. It is well written. It isn’t to long, so the readers can read it.

I suggest to shortness the abstract section. It’s too long. There are enough or adequate tables and figures.

Instead of the world “sex” is better to use “gender”

In the discussion section the authors try to explain the gender differences. In this paragraph I suggest to add this reference:

 Puca E, Pipero P, Harxhi A, Abazaj E, Gega A, Puca E, Akshija I (2018) The role of gender in the prevalence of human leptospirosis in Albania. J Infect Dev Ctries 12:150-155. doi: 10.3855/jidc.9805

 The authors are correct with citation of the study limitations.

I suggest to add and this reference in their paper

Puca E, Qato M, Pipero P, Akshija I, Kote M, Kraja D. Two cases of imported hemorrhagic fever with renal syndrome and systematic review of literature. Travel Med Infect Dis. 2019 Mar-Apr;28:86-90. doi: 10.1016/j.tmaid.2018.07.010. Epub 2018 Aug 13. PMID: 30114480.

I prefer to accept this paper with minor revision.

Reviewer #3: The manuscript presents original research on a significant topic – by authors’ account it is the largest-scale epidemiology study on hemorrhagic fever with renal syndrome reported sofar. I believe it is of interest for the readership of the PLOS Neglected Tropical Diseases. The manuscript is nicely written, statistical methods employed are state-of the art, however, the results could be presented in a more clear way, it is a bit difficult to follow. I suggest the manuscript to be accepted for publication, with some minor changes and suggestions for improvement

PLOS authors have the option to publish the peer review history of their article (what does this mean?). If published, this will include your full peer review and any attached files.

Reviewer #1: No

Reviewer #2: Yes: Edmond Puca

Reviewer #3: Yes: Maja Stanojevic

Figure Files:

Data Requirements:

Reproducibility:

References

---

## [Decision Letter · Decision Letter 1]

21 Jul 2021

Dear Mr. Sun,

We are pleased to inform you that your manuscript 'The changing epidemiology of hemorrhagic fever with renal syndrome in Southeastern China during 1963-2020: a retrospective analysis of surveillance data' has been provisionally accepted for publication in PLOS Neglected Tropical Diseases.

Best regards,

Ran Wang, Ph.D., M.D.

Associate Editor

David Harley

Deputy Editor

Reviewer's Responses to Questions

**Key Review Criteria Required for Acceptance?**

**Methods**

-Are the objectives of the study clearly articulated with a clear testable hypothesis stated?

-Is the study design appropriate to address the stated objectives?

-Is the population clearly described and appropriate for the hypothesis being tested?

-Is the sample size sufficient to ensure adequate power to address the hypothesis being tested?

-Were correct statistical analysis used to support conclusions?

-Are there concerns about ethical or regulatory requirements being met?

Reviewer #1: (No Response)

Reviewer #2: The stady is clearly articulated.

Reviewer #3: -Are the objectives of the study clearly articulated with a clear testable hypothesis stated?

yes

-Is the study design appropriate to address the stated objectives?

yes

-Is the population clearly described and appropriate for the hypothesis being tested?

yes

-Is the sample size sufficient to ensure adequate power to address the hypothesis being tested?

yes

-Were correct statistical analysis used to support conclusions?

yes

-Are there concerns about ethical or regulatory requirements being met?

no

**Results**

-Does the analysis presented match the analysis plan?

-Are the results clearly and completely presented?

-Are the figures (Tables, Images) of sufficient quality for clarity?

Reviewer #1: (No Response)

Reviewer #2: yes

Reviewer #3: Does the analysis presented match the analysis plan?

yes

-Are the results clearly and completely presented?

yes

-Are the figures (Tables, Images) of sufficient quality for clarity?

yes

**Conclusions**

-Are the conclusions supported by the data presented?

-Are the limitations of analysis clearly described?

-Do the authors discuss how these data can be helpful to advance our understanding of the topic under study?

-Is public health relevance addressed?

Reviewer #1: (No Response)

Reviewer #2: yes

Reviewer #3: -Are the conclusions supported by the data presented?

yes

-Are the limitations of analysis clearly described?

yes

-Do the authors discuss how these data can be helpful to advance our understanding of the topic under study?

yes

-Is public health relevance addressed?

yes

**Editorial and Data Presentation Modifications?**

Reviewer #1: (No Response)

Reviewer #2: Accept

Reviewer #3: none

**Summary and General Comments**

Reviewer #1: The authors have revised their manuscript according to the suggestions of the reviewer. The paper has been apparently improved. The reviewer agrees that the paper can go forward to the consideration for the publication in PLOS Neglected Tropical Diseases.

Reviewer #2: no comments

Reviewer #3: accept

PLOS authors have the option to publish the peer review history of their article (what does this mean?). If published, this will include your full peer review and any attached files.

---

## [Editor Report · Acceptance letter]

29 Jul 2021

Dear Mr. Sun,

We are delighted to inform you that your manuscript, "The changing epidemiology of hemorrhagic fever with renal syndrome in Southeastern China during 1963-2020: a retrospective analysis of surveillance data," has been formally accepted for publication in PLOS Neglected Tropical Diseases.

Best regards,

Shaden Kamhawi

co-Editor-in-Chief

Paul Brindley

co-Editor-in-Chief
